# Experimental Advances in Phase Estimation with Photonic Quantum States

**DOI:** 10.3390/e27070712

**Published:** 2025-07-01

**Authors:** Laura T. Knoll, Agustina G. Magnoni, Miguel A. Larotonda

**Affiliations:** 1CITEDEF & UNIDEF-CONICET, J.B. de La Salle 4397, Villa Martelli, Buenos Aires 1603, Argentina; magnoni.agustina@gmail.com (A.G.M.); mlarotonda@citedef.gob.ar (M.A.L.); 2Departamento de Física, FCEyN, UBA, Ciudad Universitaria, Buenos Aires 1428, Argentina

**Keywords:** quantum metrology, quantum parameter estimation, photonic quantum phase estimation

## Abstract

Photonic quantum metrology has emerged as a leading platform for quantum-enhanced precision measurements. By taking advantage of quantum resources such as entanglement, quantum metrology enables parameter estimation with sensitivities surpassing classical limits. In this review, we describe the basic tools and recent experimental progress in the determination of an optical phase with a precision that may exceed the shot-noise limit, enabled by the use of nonclassical states of light. We review the state of the art and discuss the challenges and trends in the field.

## 1. Introduction

Quantum mechanics has transformed our understanding of nature, introducing counterintuitive phenomena such as superposition and entanglement. These quantum features offer opportunities not only for foundational science but also for practical applications. Among these, quantum metrology has emerged as a powerful approach to achieving precision measurements that outperform classical strategies. In recent years, quantum technologies have reached an unprecedented pace of development, with quantum metrology playing a central role in this revolution. By designing protocols that exploit nonclassical resources, quantum metrology enables enhanced estimation of physical parameters, pushing measurement sensitivity beyond the classical bounds.

The applications of quantum metrology are vast and diverse, ranging from gravitational-wave detection [1,2] to quantum imaging of biological samples [3,4,5,6,7,8,9,10]. A particularly promising platform for implementing these advancements is photonic quantum metrology, which harnesses quantum states of light to probe physical systems [11,12,13,14]. Photons are ideal carriers of quantum information: they are robust, easily manipulated, and can be generated and detected with high precision using current technologies.

At the heart of quantum metrology lies the task of parameter estimation, where an unknown physical quantity is encoded into a quantum probe and then inferred from measurements on the output state [15]. Within this framework, phase estimation stands out as one of the most relevant problems [16]. The measurement of an optical phase is fundamental, as it can be directly related to various experimental quantities, such as distance, birefringence, angle, or sample concentration [1,2,17,18,19,20,21,22,23]. To detect a phase shift, optical interferometers are essential. The parameter is encoded during the evolution of the probe, and the information is extracted through appropriate measurements at the output.

Evaluating the uncertainty associated with parameter estimation is a central issue. The variance of an estimator quantifies the sensitivity of the measurement protocol. According to the Central Limit Theorem (CLT), repeating a measurement *M* times results in a variance scaling as ${\left(\Delta \hat{\phi }\right)}^{2}∼1/M$ defining the bound for classical strategies, known as the *shot-noise limit* (SNL). However, quantum resources such as entanglement and squeezing enable more precise estimations. Quantum metrology can reach the *Heisenberg limit* (HL), the ultimate bound set by quantum mechanics, with a variance scaling as ${\left(\Delta \hat{\phi }\right)}^{2}∼1/M^{2}$ [24,25].

Traditional phase estimation techniques rely on interferometers such as the Mach–Zehnder or Michelson interferometers fed by classical states of light, such as coherent states, which are limited by the SNL. Quantum-enhanced approaches use resources like N00N states [24,25], which offer superior phase sensitivity. Notably, recent experimental breakthroughs have demonstrated that quantum advantages can still be realized in the presence of realistic imperfections, such as photon loss or partial distinguishability [13]. This progress highlights the practical viability of quantum metrology and its transition from idealized theoretical models to robust, real-world applications.

As research advances, photonic quantum metrology is evolving from a proof-of-principle domain into a versatile tool for ultra-precise sensing, laying the groundwork for next-generation technologies in quantum communication and quantum imaging [6,7,26,27].

In this review, we explore the fundamental concepts of photonic quantum metrology, with a particular emphasis on phase estimation. Rather than attempting to provide an exhaustive overview of the entire research field, our goal is to offer a clear and accessible introduction to the core theoretical tools and experimental techniques that support this area. We focus specifically on implementations using photonic systems, which are among the most mature and versatile platforms for quantum-enhanced precision measurements. By highlighting essential principles and representative experiments, we aim to provide readers, especially those new to the field, with a solid foundation for understanding how quantum resources can be harnessed to surpass classical limits in optical metrology. For those interested in a deeper understanding, a wide collection of reviews is available in the field of photonic quantum metrology [6,12,13,14,28,29], interferometry [30,31,32] and quantum information [33,34,35], to name a few.

This review is organized as follows: in Section 2, we present the general framework for quantum parameter estimation, addressing the key concepts to evaluate the precision in the estimation process: the Cramér–Rao bound and the Fisher information, both in a classical and in a quantum scenario. We then focus on the particular problem of phase estimation in Section 3. In Section 4, we go through the current state of the art and trends in photonic implementations for phase estimation. Finally, in Section 5, we analyze the open problems and perspectives in the field.

## 2. Theoretical Tools for Parameter Estimation

The main goal of an estimation process is to obtain a numerical value (an estimate) of an unknown parameter from a set of measurement outcomes. In a general scheme, the parameter is encoded in a probe state, and information about it is retrieved after performing a measurement. We will restrict our description to quantum systems, where probe states are described by density matrices, and the encoding of the parameter is realized by a quantum operation. The general procedure for quantum parameter estimation can be described following four basic steps (sketched in Figure 1) [13]:(i)preparation of an initial quantum state $\rho^{\mathrm{in}}$, the so-called probe state;(ii)encoding of the parameter ϕ to be estimated into a quantum state $\rho^{\mathrm{out}}\left(\phi \right)=E_{\phi }\left(\rho^{\mathrm{in}}\right)$ by means of a quantum operation $E_{\phi }$ applied to the initial state;(iii)performing a measurement on the final state $\rho^{\mathrm{out}}\left(\phi \right)$ that is, in general, given by a positive-operator valued measure (POVM) $M=\{M_{x_{i}}\}$ with possible outcomes $x_{i}$;(iv)obtaining an estimator $\hat{\phi }$ from the probabilities of the outcomes that are, according to the Born rule, $P\left(x_{i}|\phi \right)=\mathrm{Tr}\left[M_{x_{i}}\rho^{\mathrm{out}}\left(\phi \right)\right]$.

An estimator is, therefore, a function of the observations, which are random variables. Different methods can be used to find an estimator, according to two fundamentally different approaches: frequentist vs. Bayesian [36]. In the frequentist approach, probabilities are viewed as the long-run frequency of events. Parameters are considered fixed but unknown quantities, and inference is based on the sampling distribution of the estimator [37]. In the Bayesian approach, probabilities are viewed as a measure of knowledge about an event. Parameters are treated as random variables with *prior* distributions that represent prior knowledge about the parameter. These distributions are updated with the observations to form posterior distributions, reflecting the updated knowledge [38].

We will focus particularly on maximum likelihood estimation (MLE), which is a fundamental frequentist method used to estimate the parameters of a model by maximizing the *likelihood function*. This approach involves identifying the values of the parameters that make the observed data more probable under the assumed model.

### 2.1. Maximum Likelihood Estimation

The likelihood function (LF) of *N* independent measurements $x_{1}\ldots x_{N}$, for a given parameter ϕ is defined as(1)$$L\left(\bar{x}|\phi \right)=\prod_{i=1}^{N}P\left(x_{i}|\phi \right),$$
where $P(x_{i}|\phi )$ is the conditional probability of obtaining the measurement $x_{i}$, given ϕ. The LF, therefore, represents the joint conditional probability of obtaining outcomes $x_{1}\ldots x_{N}$, given the parameter ϕ. The ML estimator is given by the parameter value that maximizes the likelihood function, obtained by evaluating the first and second derivatives with respect to ϕ (if L is twice differentiable):(2)$$\frac{\partial L(\bar{x}|\phi )}{\partial \phi }=0\;\mathrm{and}\;\frac{\partial^{2}L\left(\bar{x}|\phi \right)}{\partial^{2}\phi }<0.$$

It is often convenient to work with the log-likelihood function, given that both L and its logarithm attain their maxima for the same parameter value, such that(3)$$\frac{\partial \ln L(\bar{x}|\phi )}{\partial \phi }=\frac{\partial }{\partial \phi }\sum_{i=1}^{N}\ln P\left(x_{i}|\phi \right).$$

Estimators are evaluated based on several properties to ensure that they provide reliable and accurate estimates. The MLE procedure provides good estimators in the sense that these satisfy key properties:Consistency: the estimator converges (in probability) to the true value of the parameter;Unbiasedness: the mean value of the estimator’s distribution is equal to the true value of the parameter;Efficiency: the estimator has the smallest possible variance among all unbiased estimators.

As the sample size increases, the distribution of the ML estimators approaches a normal distribution, with a mean equal to the true parameter value. Given the efficiency condition, MLEs achieve the lowest possible variance among all unbiased estimators. However, there is a fundamental limit given by the Cramér–Rao bound (CRB), which provides a lower bound on the variance [39,40]. In what follows, we will first focus on the classical description of the CRB and then extend it to the quantum framework.

### 2.2. Cramér–Rao Bound and Fisher Information

The Cramér–Rao bound sets a lower bound to the variance of any estimator, given by the following fundamental inequality:(4)$${\left(\Delta \hat{\phi }\right)}^{2}\geq \frac{{\left(\frac{\partial \langle \hat{\phi }\rangle }{\partial \phi }\right)}^{2}}{F(\phi )}$$
with F(ϕ) the Fisher information (FI) given by(5)$$F\left(\phi \right)=\sum_{x}\frac{1}{L(x|\phi )}{\left(\frac{\partial L(x|\phi )}{\partial \phi }\right)}^{2}.$$

For unbiased estimators, where $\frac{\partial \langle \hat{\phi }\rangle }{\partial \phi }=1$, the CRB reduces to the inverse of the Fisher information. In addition, an estimator that saturates the CRB is said to be efficient. In the limit of a large number of measurements, the MLE produces an efficient estimator whenever there is one.

We will derive the inequality in Equation (Equation 4) following a few simple steps. We start from the definition of the mean value of the estimator, given by(6)$$\langle \hat{\phi }\rangle =\sum_{x}L\left(x|\phi \right)\hat{\phi }\left(x\right).$$

Let us now calculate the partial derivative of this expression with respect to the parameter ϕ:(7)$$\begin{array}{cc} \frac{\partial \langle \hat{\phi }\rangle }{\partial \phi }= & \sum_{x}\hat{\phi }\left(x\right)\frac{\partial L(x|\phi )}{\partial \phi }+\sum_{x}L\left(x|\phi \right)\frac{\partial \hat{\phi }\left(x\right)}{\partial \phi }=\\ = & \sum_{x}L\left(x|\phi \right)\hat{\phi }\left(x\right)\frac{\partial \ln L(x|\phi )}{\partial \phi }=\langle \hat{\phi }\left(x\right)\frac{\partial \ln L(x|\phi )}{\partial \phi }\rangle , \end{array}$$
the range of *x* is independent of ϕ, such that $\frac{\partial \hat{\phi }\left(x\right)}{\partial \phi }=0$.

On the other hand, given that $\sum_{x}L\left(x|\phi \right)=1$, when calculating the partial derivative of this expression with respect to ϕ, we obtain(8)$$0=\frac{\partial }{\partial \phi }\sum_{x}L\left(x|\phi \right)=\sum_{x}L\left(x|\phi \right)\frac{\partial \ln L(x|\phi )}{\partial \phi }=\langle \frac{\partial \ln L(x|\phi )}{\partial \phi }\rangle .$$

By multiplying Equation (Equation 8) by $\langle \hat{\phi }\rangle$ and then subtracting it from Equation (Equation 7), we obtain(9)$$\frac{\partial \langle \hat{\phi }\rangle }{\partial \phi }=\langle \left(\hat{\phi }\left(x\right)-\langle \hat{\phi }\rangle \right)\frac{\partial \ln L(x|\phi )}{\partial \phi }\rangle .$$

Finally, we take the square of this expression and apply the Cauchy–Schwartz inequality, such that(10)$${\left(\frac{\partial \langle \hat{\phi }\rangle }{\partial \phi }\right)}^{2}\leq \langle {\left(\hat{\phi }\left(x\right)-\langle \hat{\phi }\rangle \right)}^{2}\rangle \langle {\left(\frac{\partial \ln L(x|\phi )}{\partial \phi }\right)}^{2}\rangle .$$

After rearranging the terms and identifying the variance ${\left(\Delta \hat{\phi }\right)}^{2}$ and the FI, this expression is exactly Equation (Equation 4).

### 2.3. Quantum Cramér–Rao Bound and Quantum Fisher Information

From the description of parameter estimation given at the beginning of Section 2, the variance of the estimator ${\left(\Delta \hat{\phi }\right)}^{2}$ depends on the chosen probe state, the quantum operation that encodes the parameter, and the performed measurement. For fixed probe states, quantum mechanics sets an upper bound to the CRB by maximizing the Fisher information over all possible measurements. The quantum Fisher information (QFI) is then defined as(11)$$F_{Q}\left(\phi \right)=\max_{M}F\left(\phi \right).$$
which, in turn, defines the quantum Cramér–Rao bound (QCRB) such that for unbiased estimators, the following inequality holds(12)$${\left(\Delta \hat{\phi }\right)}^{2}\geq \frac{1}{F(\phi )}\geq \frac{1}{F_{Q}\left(\phi \right)}.$$

The QFI, therefore, sets the ultimate limit on precision [41,42]. To find this bound, let us express Equation (Equation 5) using the Born rule for the probabilities and assume a single repetition such that $L\left(\bar{x}|\phi \right)=P\left(x|\phi \right)=\mathrm{Tr}\left[M_{x}\rho_{\phi }\right]$. Then,(13)F(ϕ)=∑x1Tr[Mxρϕ]∂∂ϕTr[Mxρϕ]2=∑x1Tr[Mxρϕ]Tr∂ρϕ∂ϕMx2=∑x1Tr[Mxρϕ]ReTr[ρϕLϕMx]2,
where we have used the Symmetric Logarithmic Derivative (SLD) Hermitian operator $L_{\phi }$ for the last step, defined as the solution to the equation(14)$$\frac{\partial \rho_{\phi }}{\partial \phi }=\frac{\rho_{\phi }L_{\phi }+L_{\phi }\rho_{\phi }}{2}.$$

The expression in Equation (Equation 13) can be upper-bounded using the following inequalities:ReTr[ρϕLϕMx]2=|Tr[ρϕLϕMx]|2−ImTr[ρϕLϕMx]2≤|Tr[ρϕLϕMx]|2, where the equality holds if and only if $\mathrm{Tr}[\rho_{\phi }L_{\phi }M_{x}]$ is a real number ∀x:(15)ImTr[ρϕLϕMx]2=0$|\mathrm{Tr}\left[\rho_{\phi }L_{\phi }M_{x}\right]{|}^{2}\leq \mathrm{Tr}\left[\rho_{\phi }M_{x}\right]\mathrm{Tr}\left[M_{x}L_{\phi }\rho_{\phi }L_{\phi }\right]$ where we used the Cauchy–Schwarz inequality $|\mathrm{Tr}\left[A^{†}B\right]{|}^{2}\leq \mathrm{Tr}\left[A^{†}A\right]\mathrm{Tr}\left[B^{†}B\right]$, taking $A=\sqrt{\rho }\sqrt{M}$ and $B=\sqrt{\rho }L\sqrt{M}$. The equality holds if and only if(16)$$\frac{\rho M}{\mathrm{Tr}[\rho M]}=\frac{\rho LM}{\mathrm{Tr}[\rho LM]}$$

Combining the above expressions with Equation (Equation 13), we obtain(17)F(ϕ)=∑xReTr[ρϕLϕMx]2Tr[Mxρϕ]≤∑xTr[MxLρϕLϕ]=Tr[ρϕLϕ2]≡FQ(ϕ)
where, by definition, the QFI does not depend on the POVM. For the equality to hold, conditions expressed in Equations (Equation 15) and (Equation 16) must be fulfilled for a given POVM. It can be shown that an optimal measurement exists, built from the eigenstates of $L_{\phi }$ [34,43].

For the particular case of unitary transformations such that $\rho_{\phi }=e^{-i\phi H}\rho_{0}e^{i\phi H}$ (where *H* is a Hermitian operator), the QFI does not depend on the parameter to be estimated. Moreover, for initial pure sates $\rho_{0}=|\Psi_{0}\rangle \langle \Psi_{0}|$, it has a simple expression:(18)$$F_{Q}\left(\rho_{o}\right)=4{\left(\Delta H\right)}^{2}$$
with ${\left(\Delta H\right)}^{2}=\langle \Psi_{0}|H^{2}|\Psi_{0}\rangle -{\left(\langle \Psi_{0}|H|\Psi_{0}\rangle \right)}^{2}$. A complete and detailed description of QFI and its explicit forms can be found in [33,34,35].

Essentially, for a given probe state, quantum metrology defines the ultimate limit in precision, and the goal is to determine the measurement such that $F=F_{Q}$. This involves finding the POVM that ensures that the FI of the process matches the QFI associated with the given probe state.

## 3. Phase Estimation

The phase of an electromagnetic field is a relative quantity that describes the relationship between different points in the wave. A phase *shift* expresses the relative displacement between a reference and the position within the cycle of the wave. Quantum mechanics does not account for an interferometric phase Hermitian operator; hence, the task of experimentally obtaining a reliable value for the phase at the quantum level is hard and indirect and relies on estimations rather than on measurements [16,34,44]. The estimation method consists of decoding the phase from the quantum measurement of a relevant observable in a suitable setup. The quintessential experimental arrangement for this estimation is an interferometer, where the (optical or atomic) phase is encoded into a probe state, which traverses the apparatus via two or more alternate modes or paths. Upon the coherent recombination of these modes, the phase can be estimated from the measurement of the interference pattern present in the selected observable. An efficient interferometry protocol for phase estimation requires an adequate choice of several instances: the input probe state; the transformation suffered within the interferometer, the specific observable that is measured, and the estimator that, given the available resources, allows inferring the phase shift with the minimum attainable uncertainty (Figure 2). In a classical scenario, where the apparatus is probed with uncorrelated photons, this uncertainty is bounded by the SNL: ${\left(\Delta \phi_{SNL}\right)}^{2}=1/M=1/Nn$, where N is the number of independent (identical) measurements and *n* is the mean photon number of the input state on each measurement.

The SNL on the phase sensitivity of a single-port interferometer is actually a quantum limit imposed by fluctuations of the vacuum state present at the unused port of the input beamsplitter. In fact, the source of such fluctuations “has nothing to do with fluctuations in [the input photon state]; rather, it is an intrinsic property of a standard interferometer” [45]: the same limit applies to Fock states, which have zero photon number dispersion. However, the idle port of the interferometer can be transformed from foe to friend through the proper selection of the photon state injected into it: with a smart selection of both input states, the so-called Heisenberg limit can be reached. Such a limit requires the engineering of nonclassical states, as originally suggested by Caves [46], and bounds the phase fluctuations by ${\left(\Delta \phi_{HL}\right)}^{2}=1/Nn^{2}$.

### 3.1. Interferometric Setups

The Mach–Zehnder interferometer (MZI) represents the prototype two-mode interferometer. It is composed of two balanced beam splitters (equal transmission and reflection coefficients) and two mirrors arranged in such a way that a single light source is split into two coherent beams that travel different paths before being recombined (as shown in Figure 2). A phase shift ϕ may occur between the two arms, resulting in an interference pattern that depends on the relative phase difference between the two paths.

The unitary map describing the phase encoding by an MZI with input modes *a* and *b* [47,48,49,50] is given by(19)$$U_{\phi }=\exp\left(-i\phi J_{y}\right),$$
with $J_{y}=-i\left(a^{†}b-ab^{†}\right)/2$. The annihilation and creation operators $a,b,a^{†}$, and $b^{†}$ satisfy the usual bosonic commutation relations $\left[a,a^{†}\right]=\left[b,b^{†}\right]=1$.

The ubiquitous Michelson interferometer has a comparable behavior; its double-pass configuration and particular geometry have rendered it especially suitable for implementing sophisticated experiments in high-sensitivity phase estimation, as required for gravitational wave detection [1,2,46]. Also, common-path interferometers such as the displaced Sagnac, taking advantage of its intrinsic stability, have been used to achieve high-visibility interference fringes with multi-photon states [51].

### 3.2. Optimized Probe States

Clearly, at the core of the strategy behind overcoming the SNL is the preparation of specific input states that could lead to a quantum advantage in the phase estimation task. It must be kept in mind that it is fairly easy to scale up the photon number (intensity) for a classical coherent state: in the Advanced Laser Interferometer Gravitational-wave Observatory (Advanced LIGO, [1]), the optical power inside the interferometer is increased above 100 kW, which corresponds to roughly $5\times 10^{23}$ photons per second. At the shot noise limit, this implies a phase dispersion for a classical photonic state as small as $\Delta \phi \approx 10^{-12}$. Nevertheless, many applications where low light levels are mandatory, such as phase imaging of fragile biological samples and living organisms or inspection of photosensitive surfaces, can fully exploit the quantum advantage obtained by working at or close to the Heisenberg limit.

An additional word of attention is required here regarding the losses in real experimental conditions, a point that cannot be neglected when the goal is to push the precision limits of parameter estimation in phase estimation: Davidovich et al. demonstrated that, independently of the initial state of the probe, the phase uncertainty scales as 1/n (Heisenberg limit) as long as the losses within the interferometer are kept below $n^{-1}$; for larger loss, the scaling of the uncertainty recovers the $1/\sqrt{n}$ law [52]. In other words, on average, the total loss must not exceed one photon, regardless of the total photon number of the input state. Again, this condition favors the quantum phase estimation approach in applications where weak light levels are required.

#### 3.2.1. Coherent States

The classical interferometry experiment that leads to the SNL precision corresponds to a coherent state of light entering through one port of the input beamsplitter in an MZI, whereas vacuum fluctuations are present at the other port. Coherent states are defined as eigenstates of the annihilation operator $\hat{a}$ with eigenvalue α: $\hat{a}|\alpha \rangle =\alpha |\alpha \rangle$. The mean value and variance of the photon number operator $\hat{n}={\hat{a}}^{†}\hat{a}$ are given by $\langle \hat{n}\rangle =\Delta {\hat{n}}^{2}={\left|\alpha \right|}^{2}$. Then, for the input state $|\psi_{in}\rangle =|\alpha ,0\rangle$, the phase uncertainty scales as the SNL: $\Delta \phi =1/\left|\alpha \right|=1/\sqrt{\langle n\rangle }$ [13].

#### 3.2.2. Squeezed States

Several engineered (i.e., nonclassical) input states have been used to obtain a quantum advantage in the bibliography. As a natural choice, a zero dispersion state may be used instead of a coherent state, that is, an *n*-photon Fock state: $|\psi_{in}\rangle =|n,0\rangle$. However, as Caves demonstrated in [45], such an input state gives no quantum advantage since it is still affected by the vacuum state fluctuations that are present at the unused input. Indeed, in [46], he showed that a reduction in the phase dispersion could be obtained if a “squeezed vacuum” state was used at the second input: $|\psi_{in}\rangle =S_{2}\left(\xi \right)|n,0\rangle$, where *S* is the squeeze operator and ξ the complex squeeze parameter [53]. This is an improvement with respect to the SNL, although dispersion still scales with the inverse of the square root of *n*: Precision on the phase determination is bounded by $\Delta \phi_{min}=\exp\left(-\left|\xi \right|\right)/\left|\alpha \right|$. By using squeezed states as input on large Michelson interferometers, gravitational wave detectors such as LIGO, VIRGO, and GEO demonstrated the best sensitivity ever achieved, showing an improvement factor of up to 10 better than the initial classical laser-interferometer observatory [18,54,55]. Further enhancement could be observed [56] by using frequency-dependent squeezing [57,58,59,60,61].

#### 3.2.3. Twin Fock States

The class of so-called twin Fock states can be used to obtain precision approaching the Heisenberg limit: these are non-entangled product states of two Fock states, one on each input of the interferometer. When the two-photon states are indistinguishable, modal entanglement is obtained with the input beamsplitter. In particular, for the symmetric state, $|\psi_{in}\rangle =|n,n\rangle$ (even total photon number N=2n) and the “almost symmetrical” state $|\psi_{in}\rangle =|n,(n-1)\rangle$ (odd total photon number N=2n−1), the precision on the phase estimation is bounded by ${\left(2n^{2}+2n\right)}^{-1/2}$ and ${\left(2n^{2}-1\right)}^{-1/2}$, respectively [62]. Symmetric twin Fock states have been extensively investigated and experimentally implemented to obtain a quantum advantage in the phase determination, showing the above $n^{-1}$ asymptotic scaling on the precision [48,63,64,65,66,67].

#### 3.2.4. N00N States

Of special interest is the particular case of a family of states that can be engineered by exploiting photon indistinguishability: these are the so-called N00N states, prepared not as input states but as path-entangled states *inside* the Mach–Zehnder interferometer:(20)$$|\psi_{in}\rangle =1/\sqrt{2}\left(|n,0\rangle +|0,n\rangle \right).$$For *n* = 2, this state can be obtained in a deterministic and straightforward manner as the result of a Hong–Ou–Mandel (HOM) interference between two indistinguishable single photons entering separate input ports of a 50:50 beamsplitter [68]. When two identical photons enter different input ports of a balanced beamsplitter, they exit together through the same output port, canceling the coincidence probability (that is, the probability of finding a photon at each output arm), creating a *bunching effect*, as shown schematically in Figure 3.

For larger photon numbers, however, preparation is not trivial, and experimental results with such states have been obtained using post-selection at the output [22,51,69,70,71,72,73,74]. Up to date, deterministic generation of N00N states with photon number n≫1 remains an open challenge. When used as a probe for phase estimation, N00N states allow us to obtain a precision strictly limited by the Heisenberg bound: $\Delta \phi =1/n\equiv \Delta \phi_{HL}$ [75]. However, the quantum advantage in precision achievable with multi-photon N00N states is extremely sensitive to photon losses [76]. Another drawback to the use of N00N states is the requirement of photon-number-resolving detectors. Current trends in the field show a preference for alternative probing schemes to achieve metrological advantage [77,78].

## 4. Photonic Schemes for Phase Estimation: State of the Art

Traditional approaches often struggle with scalability and sensitivity to noise. In recent years, efforts have been made to design probe states to improve precision even in the presence of noise, going beyond N00N states, whose practical use is hindered by difficulties in generation and extreme sensitivity to losses [79]. Many theoretical scenarios have been studied in which quantum strategies can exceed classical limits despite noise [80,81,82,83]. These theoretical foundations and the technological advances in photon detectors [84] have enabled experiments to go beyond the SNL.

The work of Slussarenko et al. in [85] marks a pivotal advance in quantum metrology. This work presents the first experimental demonstration surpassing the shot-noise limit using photonic quantum states without any post-selection or correction for losses and inefficiencies. Previous experiments achieved sub-SNL sensitivity by correcting for system losses or discarding imperfect data. Slussarenko et al. overcame this by employing ultra-efficient photon sources and superconducting nanowire single-photon detectors (SNSPDs), enabling genuine quantum-enhanced measurements without post-processing adjustments.

In [77] the authors build upon Slussarenko et al.’s work by combining nonlinear interferometry with stimulated emission of squeezed light. They report a 5.8-fold enhancement over the SNL, surpassing ideal five-photon N00N state precision, and emphasize scalability and robustness in their approach.

The use of two-mode squeezed vacuum states [13,86] and multi-photon-subtracted twin beams [87] has also been shown to outperform traditional N00N state-based schemes. A deterministic phase estimation scheme based on a source of Gaussian squeezed vacuum states and high-efficiency homodyne detection has also been shown to overcome the performance of a pure N00N state protocol [78]. This scheme significantly surpasses the shot noise limit and even beats the conventional Heisenberg limit by a constant factor, given by the use of squeezed states without a definite number of photons.

In theory, perfect photon indistinguishability is essential for quantum-enhanced measurements. Challenging this notion, sources of partially distinguishable photons can still achieve sensitivities beyond the classical shot-noise limit (SNL) [88,89,90]. Other works have explored non-Markovian environments to preserve quantum advantages in metrology [91], using multi-photon states and multiple passes through the system under test to demonstrate scalability and robustness in quantum sensing [92], and generating entanglement with ancillary qubits to enhance the precision in phase estimation [93,94]. A recent review discusses the potential of vortex beams—beams carrying orbital angular momentum (OAM)—in quantum metrology, suggesting that their integration could lead to reduced noise and improved precision in measurements [95].

An alternative type of interferometer, theoretically introduced by Yurke et al. in 1986 as a promise for obtaining a similar performance to that of the traditional interferometers with fewer optical elements [47], replaces the passive beamsplitters with nonlinear optical devices (parametric amplifiers) which act as two-mode squeezers. These interferometers are called SU(1,1), while the standard instruments belong to the SU(2) category. The nomenclature corresponds to the names of the Special Unitary (SU) groups that mathematically describe each interferometer family. By using SU(1,1) interferometers, one can usually obtain a gain in the signal-to-noise ratio amplification, which leads to improved precision. Such an effect is accomplished via the three main characteristics of SU(1,1) interferometers that distinguish them from traditional interferometers, which are (i) the exclusive type of coherent superpositions of waves that are only allowed by second- and higher-order wave mixing in nonlinear optical devices, (ii) the ability to generate quantum entanglement—and consequently, correlations in quantum noise—that can be canceled depending on the selected interference conditions, and (iii) the immunity to losses that is guaranteed by the amplified noise levels that exceed the quantum noise level in parametric processes. The review of this class of interferometers is beyond the scope of this work, although it is worth noticing that the advent of such accessible improvement in sensitivity has led to a substantial amount of work in the last quindecennium, both from the theoretical [96,97,98,99,100,101,102,103,104,105,106,107,108,109] and the experimental [110,111,112,113,114,115,116,117,118,119,120] points of view.

The problem of phase estimation inside an optical interferometer can be mapped to a quantum circuit [121]. This approach offers high versatility, allowing the exploration of different input probe states and detection schemes. In a recent work [122] an unknown phase shift in the presence of photon loss was simulated in a quantum circuit for four different definite photon-number states of light, each containing six photons, and using photon-number-resolving detectors. This scheme relies on a Bayesian approach, as described in [123].

Given that optical sensors based on quantum states of light are susceptible to noise and imperfections that affect their performance, the estimation of multiple parameters is relevant for a wide range of sensors. The simultaneous estimation of several parameters in a single experiment opens the door to more complex measurement devices, which aim to estimate multiple parameters related to the object or sample of interest [29,124,125,126,127,128,129,130]. The optimal strategy entails a trade-off between the achievable precisions of the various parameters. This trade-off requires an optimization strategy to determine both the best quantum state and the optimal measurement that allows reaching the ultimate precision limit in multiparameter quantum metrology.

Recently, in the Noisy Intermediate-Scale Quantum (NISQ) era of quantum computing [131], variational quantum algorithms (VQAs) [132] have emerged as a promising solution to noise sensitivity, offering flexibility and adaptability in optimizing quantum circuits for specific tasks. For the case of single-parameter metrology, VQAs have been used to optimize quantum probes [133,134,135]. Meyer et al. [136] introduced a variational toolbox for quantum multiparameter estimation, laying the theoretical foundation for the application of VQAs in metrology. Recent works have explored deep reinforcement learning for quantum multiparameter estimation, highlighting the potential of machine learning techniques to optimize quantum sensors [137]. Hybrid quantum-classical frameworks have also been introduced [138], as well as protocols combining quantum optical nonlinearities with VQAs [139], aimed at enhancing multiparameter quantum metrology and integrating variational quantum algorithms with photonic systems.

## 5. Conclusions and Perspectives

The recent experimental advances in photonic quantum metrology highlight both the maturity of current photonic platforms and the growing feasibility of real-world quantum sensors.

Despite these advances, several limitations remain that must be addressed before photonic quantum metrology can be widely used in practical technologies. A primary challenge is scalability: generating and maintaining highly entangled multi-photon states remains technically demanding, particularly as the photon number increases [140]. Photon loss, mode mismatch, and detector inefficiencies continue to affect performance in larger-scale implementations. Moreover, many protocols still rely on post-selection or operate under idealized assumptions that limit their applicability in noisy environments. Recently, the emerging field of non-Hermitian quantum metrology has achieved promising progress in realizing non-Hermitian systems at the quantum level for metrology and quantum estimation tasks [141,142,143,144].

Another challenge lies in the robustness and adaptability of quantum metrological schemes. While quantum advantage has been shown in specific, controlled settings, generalizing these results to multiparameter estimation, real-time sensing, or distributed quantum networks remains an open problem. Additionally, quantum resource quantification and the identification of the optimal trade-off between metrological precision and experimental complexity are still active areas of research.

Advances in integrated photonics and detector technologies [145,146,147] and in machine-learning-assisted protocols [136,148,149,150] are likely to mitigate many current limitations. In the near term, we can expect improvements in the implementation of quantum-enhanced metrological protocols using imperfect but realistic resources. Photonic quantum metrology will continue to contribute to the development of quantum technologies, with increasing applications to quantum communication, computation, and precision navigation. The integration of metrology with emerging quantum networks and the transition toward fully autonomous, adaptive quantum sensors represent exciting directions for future research and development.

## Figures and Tables

**Figure 1 entropy-27-00712-f001:**
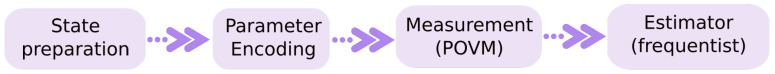
Quantum estimation process. An initial probe state is prepared and sent to interact with a quantum operation that encodes the parameter to be estimated. Measurements are then performed on the output state, and an estimator is inferred from the measured probability distributions.

**Figure 2 entropy-27-00712-f002:**
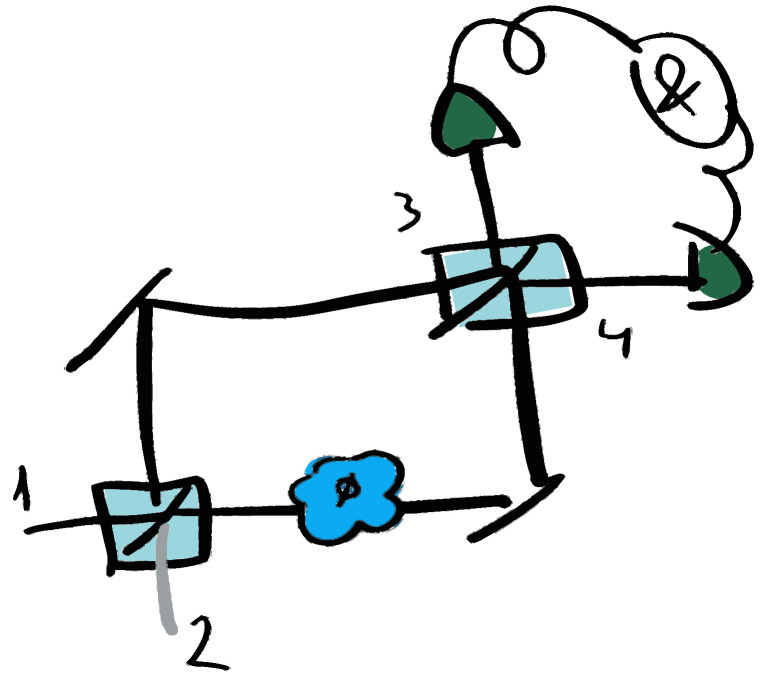
A schematic representation of a Mach–Zehnder interferometer for phase estimation in a classic framework. A probe state is sent to the apparatus through inputs 1 and 2, where a two-mode superposition state is created at the first beamsplitter. This state interacts with a phase encoding mechanism. The phase can be estimated from measurements of a specific observable at outputs 3 and 4 after the coherent recombination of the two modes at the output beamsplitter.

**Figure 3 entropy-27-00712-f003:**
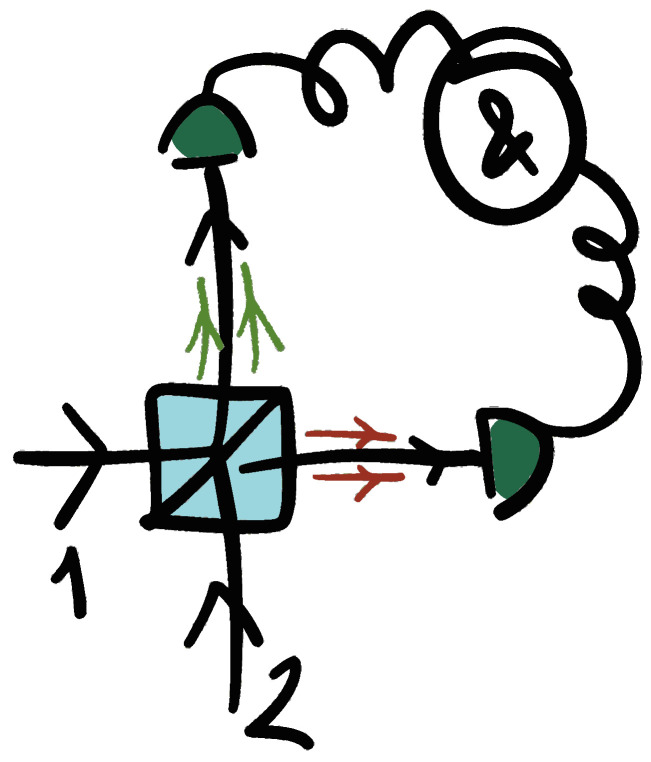
Scheme for HOM interference. Two identical single photons enter ports 1 and 2 of a beamsplitter and exit *bunched* together through one or the other output port, illustrated by the double green and red arrows.

## Data Availability

Not applicable.

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
