# Peer review of "Experimental Advances in Phase Estimation with Photonic Quantum States"

_entropy, 2025, doi:10.3390/e27070712_

Round 1

Reviewer 1 Report

Comments and Suggestions for Authors

In this work, the authors gave an update and comprehensive review on the current state of phase estimation with photonic quantum states. In particular,  this review starts with a general framework in Section 2 for quantum parameter estimation. Then,  particular problem of phase estimation  is given in Section 3,  for Mach-Zehnder interferometer and  optimized probe state. Photonic schemes for phase estimation are illustrated in Section 4.

Some comments are listed below:

1, For the phase estimation, only Mach-Zehnder interferometer is mentioned in Section 3.1. To enhance the readability as a Review article, different types of interferometry must be added, which in particular are also important for the photonic quantum states.

2, In the content of gravitational wave detectors, nowadays, the successful implementation with squeezing states must also be added, including

L. McCuller, C. Whittle, D. Ganapathy et al., Frequency-dependent squeezing for Advanced LIGO, Phys. Rev. Lett. 124, 171102 (2020).
Y. Zhao, N. Aritomi, E. Capocasa et al., Frequency-dependent squeezed vacuum source for broadband quantum noise reduction in advanced gravitational-wave detectors, Phys. Rev. Lett. 124, 171101 (2020).

D. Ganapathy, et al., Broadband Quantum Enhancement of the LIGO Detectors with Frequency-Dependent Squeezing, Phys. Rev. X 13, 041021 (2023).

Overall, the review not only summarizes the relevant development, but also emphasizes the potential applications. I am happy to give my recommendation for publication in Entropy. 

Author Response

We wish to thank the Referees for the time spent reviewing our manuscript and for their useful comments and suggestions, that allowed us to improve the paper.

A detailed list of our revisions is reported in the following, together with a point-by-point reply to all Referees' comments.

The changes made in the revised manuscript are written in blue in order to track them.

Comments 1: For the phase estimation, only Mach-Zehnder interferometer is mentioned in Section 3.1. To enhance the readability as a Review article, different types of interferometry must be added, which in particular are also important for the photonic quantum states.

Response 1: We agree with the reviewers' comment regarding other interferometric setups. We have therefore changed the name of section 3.1 to "Interferometric setups" and added a paragraph at the end mentioning the Michelson interferometer and its use for gravitational wave detection, as well as the displaced Sagnac interferometer.

Comments 2: In the content of gravitational wave detectors, nowadays, the successful implementation with squeezing states must also be added, including
-L. McCuller, C. Whittle, D. Ganapathy et al., Frequency-dependent squeezing for Advanced LIGO, Phys. Rev. Lett. 124, 171102 (2020).
-Y. Zhao, N. Aritomi, E. Capocasa et al., Frequency-dependent squeezed vacuum source for broadband quantum noise reduction in advanced gravitational-wave detectors, Phys. Rev. Lett. 124, 171101 (2020).
-D. Ganapathy, et al., Broadband Quantum Enhancement of the LIGO Detectors with Frequency-Dependent Squeezing, Phys. Rev. X 13, 041021 (2023).

Response 2: We thank the Reviewer for pointing this out. We agree that frequency-dependent squeezed states must be included in our Review. Therefore, we have added a description in Section 3.2.2 ("Squeezed states"), lines 263-268. We have included the references suggested by the Reviewer [59,60,62] as well as the following ones:
[55] Tse, M., et al. Quantum-enhanced advanced LIGO detectors in the era of gravitational-wave astronomy. Physical Review Letters 2019, 499
123, 231107.
[56] Acernese, F., et al. Increasing the astrophysical reach of the advanced virgo detector via the application of squeezed vacuum states of light. Physical Review Letters 2019, 123, 231108.
[57] Capote E., et al. Advanced LIGO detector performance in the fourth observing run. Physical Review D 2025, 111, 062002.
[58] Evans, M., et. al. Realistic filter cavities for advanced gravitational wave detectors. Physical Review D—Particles, Fields, Gravitation, and Cosmology 2013, 88, 022002. 
[61] Polini, E. Broadband quantum noise reduction via frequency dependent squeezing for Advanced Virgo Plus. Physica Scripta 2021, 513
96, 084003. 

Reviewer 2 Report

Comments and Suggestions for Authors

Comments included in the attached file

Author Response

We wish to thank the Referees for the time spent reviewing our manuscript and for their useful comments and suggestions, that allowed us to improve the paper.

A detailed list of our revisions is reported in the following, together with a point-by-point reply to all Referees' comments.

The changes made in the revised manuscript are written in blue in order to track them.

Comments 1: As the authors
classify it as a review, it would be interesting to also include some late approaches, such as the
estimation of the unknown phase shift inside a Mach–Zehnder interferometer in the presence of
photon loss based on simulating quantum circuits. The mechanism employs the Bayesian approach
in which likelihood functions are the outcome of simulating the corresponding quantum circuits
[see Najafi et al, Sci. Rep. 13 (2023) or Zhou et al, Quant. Information Processing 23 (2024)].

Response 1: We thank the Reviewer for the observation. We have now included a brief discussion on the simulation of phase estimation using quantum circuits in Section 4, lines 354-360. We have added the suggested references as well as the following one:
[122] Lee, H. et al. A quantum Rosetta stone for interferometry. Journal of Modern Optics 2002, 49, 2325–2338.

Comments 2: I also consider it is beneficial for the paper that the authors also detail on the measurement
of the optical phase enabled by optical interferometry, where measured phase error is constrained
by the Heisenberg limit. Phase estimation at the Heisenberg limit has been generally achieved
based on highly complex N00N states of light. A deterministic phase estimation scheme is
suggested is suggested in PRL 130 by Nielsen et al, which employs a source of Gaussian squeezed
vacuum states and high-efficiency homodyne detection. The outcome of the scheme lies in phase
estimates with extreme sensitivity, which outperforms both the shot noise and the Heisenberg
limit, as well as the performance of a pure N00N state protocol.

Response 2: We agree with the Reviewer on the significance of the work done by Nielsen et al (PRL 130, 123603 - 2023). This work was cited in our manuscript, but no description of the experiment was made. In the revised version, details of this paper (ref [79]) can be found in Section 4, lines 320-324.